

# A new parallel multi-objective Harris hawk algorithm for predicting the mortality of COVID-19 patients

Tansel Dokeroglu

Cankaya University, Software Engineering Department, Ankara, Turkey

## ABSTRACT

Harris' Hawk Optimization (HHO) is a novel metaheuristic inspired by the collective hunting behaviors of hawks. This technique employs the flight patterns of hawks to produce (near)-optimal solutions, enhanced with feature selection, for challenging classification problems. In this study, we propose a new parallel multi-objective HHO algorithm for predicting the mortality risk of COVID-19 patients based on their symptoms. There are two objectives in this optimization problem: to reduce the number of features while increasing the accuracy of the predictions. We conduct comprehensive experiments on a recent real-world COVID-19 dataset from Kaggle. An augmented version of the COVID-19 dataset is also generated and experimentally shown to improve the quality of the solutions. Significant improvements are observed compared to existing state-of-the-art metaheuristic wrapper algorithms. We report better classification results with feature selection than when using the entire set of features. During experiments, a 98.15% prediction accuracy with a 45% reduction is achieved in the number of features. We successfully obtained new best solutions for this COVID-19 dataset.

## INTRODUCTION

The coronavirus disease 2019 (COVID-19), caused by the severe acute respiratory syndrome coronavirus 2 (SARS-CoV-2), has become a significant concern for countries worldwide (*Albahri et al., 2020*). The coronavirus disease pandemic has devastated the worldwide population and overwhelmed advanced healthcare systems. As of April 2023, there have been approximately 686 million COVID-19 cases and 6,862,569 deaths (www.worldometers.info). Given the virus's potential threat to public health, it is crucial to identify indicators that can serve as reliable predictors of COVID-19 patients' clinical outcomes and classify patients' severity as soon as possible (*Lai et al., 2020*; *Dokeroglu, Deniz & Kiziloz, 2021*). However, this field's lack of sophisticated machine-learning applications makes it computationally challenging. Considering their past achievements, metaheuristics wrapper classification algorithms are evaluated as effective techniques in this domain.

Harris' Hawks Optimization (HHO) is a unique metaheuristic inspired by the collective behaviour of hawks (*Heidari et al., 2019*; *Jangir, Heidari & Chen, 2021*). These avians use

Corresponding author
Tansel Dokeroglu,
tanseldoker@gmail.com

clever strategies, such as surprise pounce (seven kills), to grab their prey based on the escape tendencies of the victim. The HHO metaheuristic mimics the hawks' hunting habits to find the (near)-optimal solutions to the NP-Hard problems. In this study, we propose a novel parallel multi-objective HHO method (PHHO-KNN) using K-nearest neighbour (KNN) classifiers for predicting the mortality of COVID-19 patients according to their findings (features). *Liu et al. (2023)* proposed a new hybrid HHO algorithm for tumour feature gene selection. They evaluated each gene through the variance filter and selected the best feature genes. Later, they used the HHO metaheuristic to select the best subset of genes. The experiments showed 100% classification for gastric cancer and an average accuracy of 95.33% for other cancers. *Too, Abdullah & Mohd Saad (2019)* proposed a binary HHO (BHHO) for feature subset selection. The BHHO algorithm used S and V-shaped transfer functions to obtain selected features. The superiority of the BHHO is verified on these datasets.

Metaheuristic algorithms need to calculate numerous candidate solutions to find the best result. Therefore, it becomes a challenging problem due to the size of the problem space. Parallel metaheuristics can be extremely effective tools when evaluated from this perspective. They can compute many more fitness calculations than their single-processor counterparts and obtain better results in shorter execution times. The diversity provided by numerous separate populations in the memory of different processors is another advantage of these algorithms. Without getting stuck with local optima, they can explore and exploit the problem space more effectively (*Alba, Luque & Nesmachnow, 2013*). The algorithm we propose also enhances the accuracy of predictions using feature selection and augmented data. Feature selection is a critical field of machine learning (*Dokeroglu, Deniz & Kiziloz, 2022*). Data analysis, information retrieval, classification, and data mining heavily rely on feature selection. It reduces the number of features by eliminating noisy, irrelevant, or redundant data. This technique is an inevitable part of big data processing, which has become one of the most important problems in our world (*Bolón-Canedo, Sánchez-Maroño & Alonso-Betanzos, 2015*). The KNN classification algorithm (*Rajammal et al., 2022*) is used as a classifier for each candidate solution generated by the HHO layer of our proposed parallel algorithm.

With this study, we aimed to investigate the potential of obtaining better results using a newly developed metaheuristic algorithm, HHO. There always exist new opportunities of obtaining better results in the field of combinatorial optimization as described in the no free lunch theorem (*Wolpert & Macready, 1997*). The Slime Mould Algorithm (SMA) (*Li et al., 2020*), Runge Kutta Method (RUN) (*Ahmadianfar et al., 2021*), Colony Predation Algorithm (CPA) (*Tu et al., 2021*), Weighted Mean of Vectors (INFO) (*Ahmadianfar et al., 2022*) and Rime Optimization Algorithm (RIME) (*Su et al., 2023*) are some of the most recent metaheuristics. The HHO algorithm is one of these newly proposed metaheuristic algorithms that have achieved exceptional results on the problems it has been applied to.

In some recent metaheuristic studies, a hybrid feature selection method based on Butterfly optimization algorithm (BOA) and Ant Lion optimizer (ALO) is proposed by *Thawkar et al. (2021)* to detect breast cancer using mammogram images, which

outperforms BOA and ALO in terms of accuracy, sensitivity, and specificity. A modified whale optimization algorithm (mWOAPR) is proposed by *Chakraborty et al. (2021)* to improve the efficiency of diagnosing the severity of COVID-19 using chest X-ray images. The method outperforms basic and modified metaheuristic algorithms, demonstrating its potential for accurate diagnosis. A new model for skin lesion classification as normal or melanoma is proposed, overcoming severe class imbalance in the ISIC 2020 dataset using a random over-sampling method followed by data augmentation. A hybrid convolutional neural network architecture and bald eagle search optimization achieved high accuracy, specificity, sensitivity, and F-score, outperforming VGG19, GoogleNet, and ResNet50 models (*Sayed, Soliman & Hassanien, 2021*). An improved version of the Whale Optimization Algorithm (WOA) called QGBWOA, is proposed to introduce quasi-opposition-based learning and Gaussian barebone mechanism to enhance its ability to search for optimal solutions. The performance of QGBWOA is tested on CEC 2014 and CEC 2020 test problems and compared to other algorithms, demonstrating improved convergence accuracy and speed (*Xing et al., 2023*). *Piri & Mohapatra (2021)* presented a new technique, Multi-Objective Quadratic Binary Harris Hawk Optimization (MOQBHHO), for feature selection (FS) in classification problems. The MOQBHHO method uses a K-nearest neighbor (KNN) classifier and crowding distance value as additional criteria to find optimal feature subsets. The proposed approach is evaluated on 12 medical datasets and compared with other FS methods.

The HHO algorithm was selected due to its reported high performance on other optimization problems. Using a multiobjective approach, we aimed to observe its performance on the COVID-19 classification problem. An augmented version of the COVID-19 dataset was also prepared, and the HHO algorithm was applied to this dataset. By evaluating the performance of HHO on feature selection and designing it as a multi-objective algorithm that aims to achieve the minimum number of features with good prediction, we can effectively increase the accuracy and scalability of the algorithm on big data. Additionally, we investigate the effectiveness of augmented data, a technique that has recently been shown to provide better results in classification accuracy. To our knowledge, no previous study has examined these techniques together on recent COVID-19 datasets. This motivated us to conduct comprehensive experiments on a recent real-world COVID-19 dataset from Kaggle. We generated an augmented version of the dataset to improve the quality of our solutions. Our experiments reveal significant improvements compared to state-of-the-art metaheuristic wrapper algorithms, and we achieved new best solutions for the COVID-19 dataset.

Our contributions to this study can be summarized as follows:

- A new parallel multi-objective HHO algorithm (PHHO-KNN) is proposed to classify COVID-19 patients.
- The results are improved using the augmented data version of the COVID-19 patients.
- Feature selection is applied to augmented COVID-19 data for the first time.

- New best accuracy values are obtained with fewer features for the first time in literature.
- Our proposed algorithm (PHHO-KNN) is the best-performing algorithm in the literature.

The "Related work" of our article reviews related studies of machine learning approaches for the COVID-19 disease. The "Proposed parallel multi-objective harris hawk algorithm" gives details about our proposed parallel algorithm, PHHO-KNN. In the experimental setup and evaluation of results section, we define our experiments, software, datasets (including our proposed augmented dataset), hardware, and comparisons with state-of-the-art algorithms. In the last section, we give our concluding remarks and future work.

## RELATED WORK

In this part, we give information about the recent state-of-the-art COVID-19 studies related to our work. *Deniz et al. (2022)* proposed a multi-threaded evolutionary feature subset selection method using extreme learning machines (MG-ELM) to classify COVID-19 patients. They conduct experiments on the Kaggle dataset using feature selection. The results are competitive with other state-of-the-art algorithms. *Dokeroglu, Deniz & Kiziloz (2021)* applied features selection and increased the accuracy level of predictions. They proposed new besiege and perching operators for the multi-objective classification problems. They conducted comprehensive experiments on UCI Machine Learning Repository. They applied the proposed algorithm to the Kaggle COVID-19 dataset. They report significant improvements compared to state-of-the-art metaheuristics. *Dokeroglu et al. (2019)* distinguished new and outstanding metaheuristics of the last two decades. They summarize the foundations of metaheuristics, recent trends, hybrid algorithms, recent problems, parallel metaheuristics, and new opportunities. *Dokeroglu & Sevinc (2019)* proposed a Parallel Genetic ELM (IPE-ELM) for data classification. The IPE-ELM uses feature selection, parallel fitness computation, and parameter tuning of hidden neuron layers. The IPE-ELM is tested using UCI benchmark datasets. This algorithm has similar ideas to our proposed algorithm. *Xue, Yao & Wu (2018)* presented a hybrid genetic algorithm and ELM (HGEFS). They proposed a novel mechanism to maintain diversity. Comparisons are carried out with benchmark datasets. The HGEFS outperforms other algorithms.

*Kiziloz et al. (2018)* proposed new multiobjective Teaching Learning Based Optimization (TLBO) techniques for the selection of features. They utilize the algorithm-specific parameterless concept of TLBO. Improvements are observed compared to NSGA-II, PSO, and Tabu Search algorithms. *Dhamodharavadhani, Rathipriya & Chatterjee (2020)* studied probabilistic models for the mortality prediction of COVID-19 patients. The results showed that the probabilistic neural network performs better than other models. *Cantú-Paz (1998)* presented a survey on parallel genetic algorithms (PGAs). They prepared a taxonomy of techniques and showed case studies. They also described the most significant issues in modeling and designing multi-population parallel GAs.

*Bullock et al. (2020)* presented an overview of Machine Learning studies to tackle many aspects of the COVID-19 crisis. We have identified applications and challenges posed by COVID-19. They reviewed datasets, tools, and resources needed to facilitate AI research. *Chen et al. (2020)* aim to develop a combination of feature selection algorithms (filter, wrapper, or embedded techniques). The experimental results showed that combining filter and wrapper algorithms is a better choice. *Huang, Ding & Zhou (2010)* studied ELM for classification and verified that ELM tends to perform better than SVM. The ELM is outstanding with its parameter settings. *Huang et al. (2011)* showed that the least square SVM and proximal SVM could be a unified framework, and other regularization algorithms referred to as ELM can be built. *Irshad, Yin & Zhang (2021)* proposed a Particle Swarm Optimization algorithm (PSO) to obtain the optimal subset of retinal vessel features. They developed an objective function and compared the classification accuracy with the state-of-the-art approaches. The proposed algorithm is validated using a special dataset.

*Iwendi et al. (2020)* proposed a Random Forest model supported AdaBoost algorithm to predict the severity of the patients. The authors report 94% accuracy and a 0.86 F1 Score. There is a positive relation between the deaths and gender of the patients (in patients between the ages of 20 and 70). *Kashef & Nezamabadi-pour (2015)* introduced an Ant Colony Optimization (ABACO) feature selection algorithm. In this model, features are graph nodes and connected to other features. The result verified that the algorithm is good for feature selection. *Mydukuri et al. (2022)* proposed Gaussian neuro-fuzzy multi-layered data classification (LSRGNFM-LDC) method. The algorithm performs good prediction results on the 2019 COVID-19 dataset. *Rasheed et al. (2020)* surveyed state-of-the-art AI algorithms applied to the context of COVID-19 for mortality rates. *Shaban et al. (2020)* proposed a new hybrid feature selection method with KNN classifiers. Experiments showed that the strategy outperforms existing techniques.

*Too & Mirjalili (2021)* developed a novel Dragonfly Algorithm (HLBDA) for feature selection and data classification. The algorithm is applied to a COVID-19 dataset. The results demonstrated that the HLBDA has outstanding accuracy and fewer features. *Umarani & Subathra (2020)* developed research using machine learning and presented the application of data mining for COVID-19. *Wu et al. (2020)* developed a model for the severity and triage of COVID-19 patients. *Yu, Beam & Kohane (2018)* outlined recent AI studies and summarized the healthcare applications in AI. *Pizarro-Pennarolli et al. (2021)* studied the impact of COVID-19 on daily activities they prepared a review. They studied post-infection cases of COVID-19 patients. *Wang et al. (2020)* proposed a random forest technique for the severity prediction of COVID-19 patients. They applied a recursive feature elimination algorithm. *Sheykhivand et al. (2021)* introduced a new algorithm for identifying COVID-19 using a deep neural network. Generative adversarial networks are used with LSTM networks. They achieved 90% accuracy.

*Shukla (2021)* introduced a new gene selection technique with TLBO for cancer prediction. Experiments verified that the method could significantly remove the irrelevant genes and outperform the wrapper algorithms. *Sun et al. (2020)* proposed a feature selection with deep forest to classify COVID-19 chest tomography images. They use a deep

forest model to learn a high-level model of features. Experimental results with COVID-19 datasets showed that the proposed algorithm achieves high performance in COVID-19. *Tayarani (2020)* prepared an AI survey on the applications of COVID-19. *Yu et al. (2020)* proposed a genetic algorithm for the admission of vital cases. Several risks were defined and they developed neural network models using Genetic Algorithms. The model outperformed generalized linear models.

## PROPOSED PARALLEL MULTI-OBJECTIVE HARRIS HAWK ALGORITHM

Parallelization is particularly important for wrapper algorithms for feature selection (*Alba, Luque & Nesmachnow, 2013*). Wrapper algorithms evaluate the fitness of a subset of features by training a machine learning model on the subset of features and assessing its performance. The wrapper algorithm searches for the optimal subset of features by iteratively evaluating different subsets and selecting the subset that yields the best performance. Parallelization can be used to speed up the evaluation process by allowing the wrapper algorithm to evaluate multiple subsets of features concurrently. This can be achieved by using multiple processors to simultaneously evaluate the fitness of different subsets of features. This approach can significantly reduce the time taken to identify the optimal subset of features, particularly when dealing with large datasets with many features. Additionally, parallelization can enable wrapper algorithms to scale to larger datasets and higher dimensional feature spaces. As the number of features and instances in a dataset increase, the time required to evaluate all possible subsets of features grows exponentially. Parallelization can help to mitigate this problem by allowing the wrapper algorithm to evaluate subsets of features concurrently. Furthermore, the parallelization of wrapper algorithms for feature selection can improve the robustness of the optimization process. By evaluating multiple subsets of features concurrently, the algorithm can explore different regions of the search space in parallel, reducing the risk of getting stuck in local optima. In summary, parallelization is crucial for wrapper algorithms for feature selection. It can improve the efficiency, scalability, and robustness of the optimization process, enabling these algorithms to identify the optimal subset of features more quickly and accurately.

Our proposed PHHO-KNN algorithm has two layers, each with a distinct task. In the first layer, we employ HHO to perform feature selection, while KNN is used to classify the patients. HHO is population-based and gradient-free and simulates the cooperative hunting patterns of hawks attacking their prey from different directions, with the aim of exhausting it and making it more vulnerable. The HHO phases are inspired by the jumps and attack strategies of the Hawks. Figure 1 illustrates the exploration and exploitation steps of HHO based on the energy and ($q$) values. To provide more details on our algorithm, we present the PHHO-KNN algorithm in Algorithm 1.

To find prey, hawks in our proposed algorithm utilize a perching strategy where they perch in trees while observing the field. The location of other hawks and the prey determines where they perch, with the condition that if the value of $q < 0.5$, perching is based on those locations; otherwise, it is random. To facilitate exploration, we have

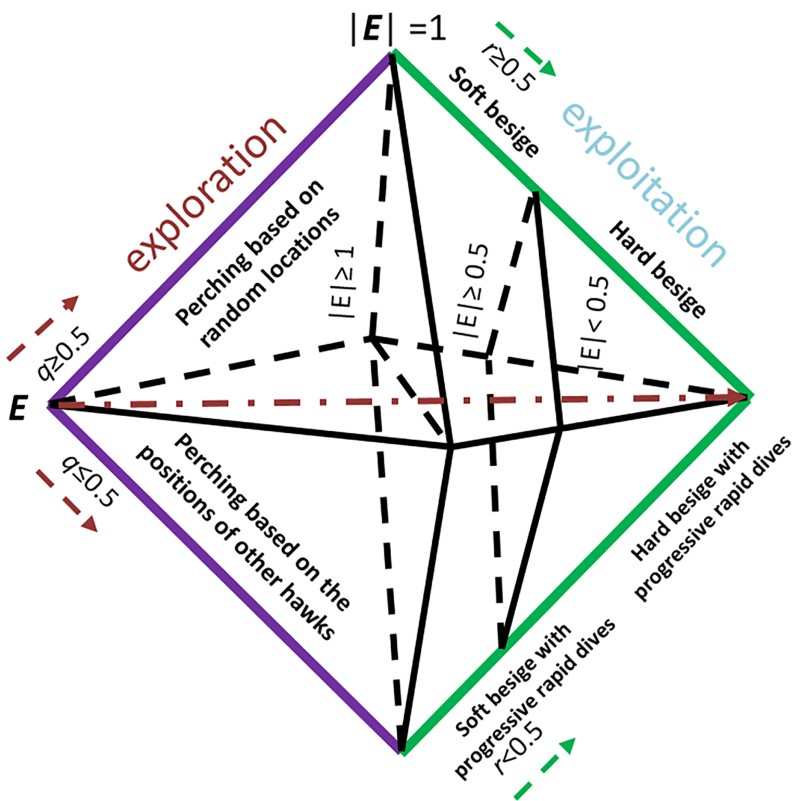

**Figure 1** **The exploitation and exploration steps are decided according to the rabbit's energy level $E$, ($q$ and $r$ are random values).**

proposed two operators in this study. The first operator, Exploration_1, randomly selects hawk_1 and hawk_2; some of hawk_1's features are copied to hawk_2. The second operator, Exploration_2, copies the selected subset of features of the best solution (hawk) to the current solution (hawk). The resulting candidate is then added to the population. The choice between exploration or exploitation is determined by the prey's escaping energy value ($E$), which is given below:

$$E = 2E_0\left(1 - \frac{t}{T}\right) \tag{1}$$

The initial energy of the prey in our proposed algorithm is denoted by $E_0$, and the total number of iterations is represented by $T$. The value of $E_0$ is between ($-1$, $1$) and increases as the prey becomes stronger. Conversely, if $E_0$ decreases, the mobility of the prey reduces. As the number of iterations increases, the value of $E$ decreases. The HHO algorithm performs exploration when $|E| \geq 1$, and exploitation when $|E| < 1$.

In our proposed algorithm, the hawks can besiege the prey by encircling it, with the type of besiege executed determined by the rabbit's energy level. Soft besiege is executed when $|E| \geq 0.5$, while hard besiege is executed when $|E| < 0.5$. If $r \geq 0.5$ and $|E| \geq 0.5$ then the energy level of the rabbit is good, and it allows it to jump to get rid of hawks where $r$ is a randomly selected value. The hawks can surround the prey and use surprising techniques

| | **Algorithm 1**: Parallel multi-objective HHO algorithm, PHHO-KNN. |
|---|---|
| 1 | **if** *(This is the master processor)* **then** |
| 2 |     **while** *(i++ < #slaves)* **do** |
| 3 |         Receive solutions from slave_i(); |
| 4 |         Update the set of best solutions(); |
| 5 | **if** *(This is a slave processor)* **then** |
| 6 |     Generate initial population $X_i (i = 0, 1, \ldots, N - 1)$ |
| 7 |     **while** *(i < iter)* **do** |
| 8 |         Decide the best location |
| 9 |         Set the location of the rabbit $(X_{rabbit})$ |
| 10 |         **for** *(each hawk $(X_i)$)* **do** |
| 11 |             Set E value of the prey |
| 12 |             **if** $(E \geq 1)$ **then** |
| 13 |                 // Start Performing Exploration |
| 14 |             **else** |
| 15 |                 // Start Performing Exploitation |
| 16 |                 **if** $(E \geq 0.5)$ **then** |
| 17 |                     Start Performing *soft besiege* |
| 18 |                 **else** |
| 19 |                     Start Performing *hard besiege* |
| 20 |         Calculate the accuracy values using KNN. |
| 21 |         Insert new and better solutions into the population. |
| 22 |     Send Results ← to the master node |

by creating actions based on a random number *J*. Additionally, the rabbit's features are inserted into the current solution. On the other hand, if $r \geq 0.5$ and $|E| < 0.5$, the prey performs jumps, and the location of the hawk is updated using the following equation:

$$X(t + 1) = X_{rabbit}(t) - E|\Delta X(t)| \qquad (2)$$

At iteration *t* in our proposed algorithm, $\Delta X(t)$ represents the difference between the rabbit and the hawk. We copy a single feature (dimension) of the prey (as shown in Fig. 2).

In our proposed algorithm, the rabbit can survive attacks when the value of $r < 0.5$ and $|E| \geq 0.5$, and we introduce a new operator with a perturbation value. According to the E value, a subset of features is copied to the recent solution (hawk). On the other hand (the values of $|E| < 0.5$ and $r < 0.5$) the hawks only reduce the distance between them and their prey. A minimum number of features are used to avoid higher perturbation levels. Our proposed algorithm, presented in Algorithm 1, does not allow redundant candidate solutions.

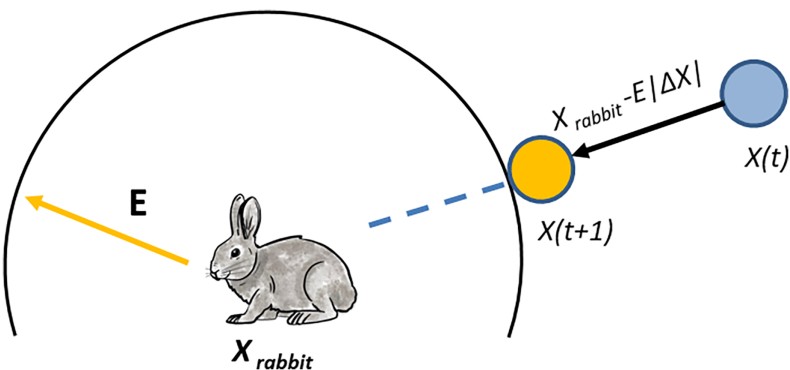

**Figure 2** **The positions of the vectors in hard besiege.**

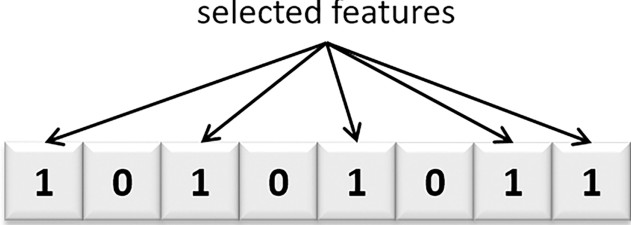

**Figure 3** **A candidate solution of our proposed algorithm.**

The PHHO-KNN algorithm proposed in this study employs a master-slave architecture. Each slave node/processor generates a distinct population, working independently of the others. Generations are performed on local populations by each slave node. There is no exchange of individuals between populations. Once the generations are completed, the slave nodes send their populations to the master node. The master node receives the solutions, eliminates similar solutions, and selects the top 30 solutions to construct a Pareto curve. Figure 3 represents a solution where the selected features are represented with value one. Additionally, the flowchart of our proposed PHHO-KNN algorithm is depicted in Fig. 4.

Our proposed algorithm's time complexity is mainly determined by the number of iterations (itr) and the population size (p). Each fitness calculation requires five-fold cross-validation of the KNN algorithm on the chosen subset of features, resulting in a total of (itr · p) fitness calculations. As a result, we account for this expense in our overall complexity. The KNN classification algorithm has a complexity of $O(n \cdot d)$, where n is the size of instances (dataset) in the training set and d is the size of features. As a result, the complexity of our proposed algorithm is $O(itr \cdot p \cdot n \cdot d)$. This is a polynomial time execution algorithm.

## K-nearest neighbours classifier (KNN)

Our algorithm uses the K-nearest neighbours (KNN) classifier, known for its high performance in recent classification studies (*Rajammal et al., 2022*). KNN is a non-parametric technique used for supervised learning (*Cunningham & Delany, 2021*; *Guo*

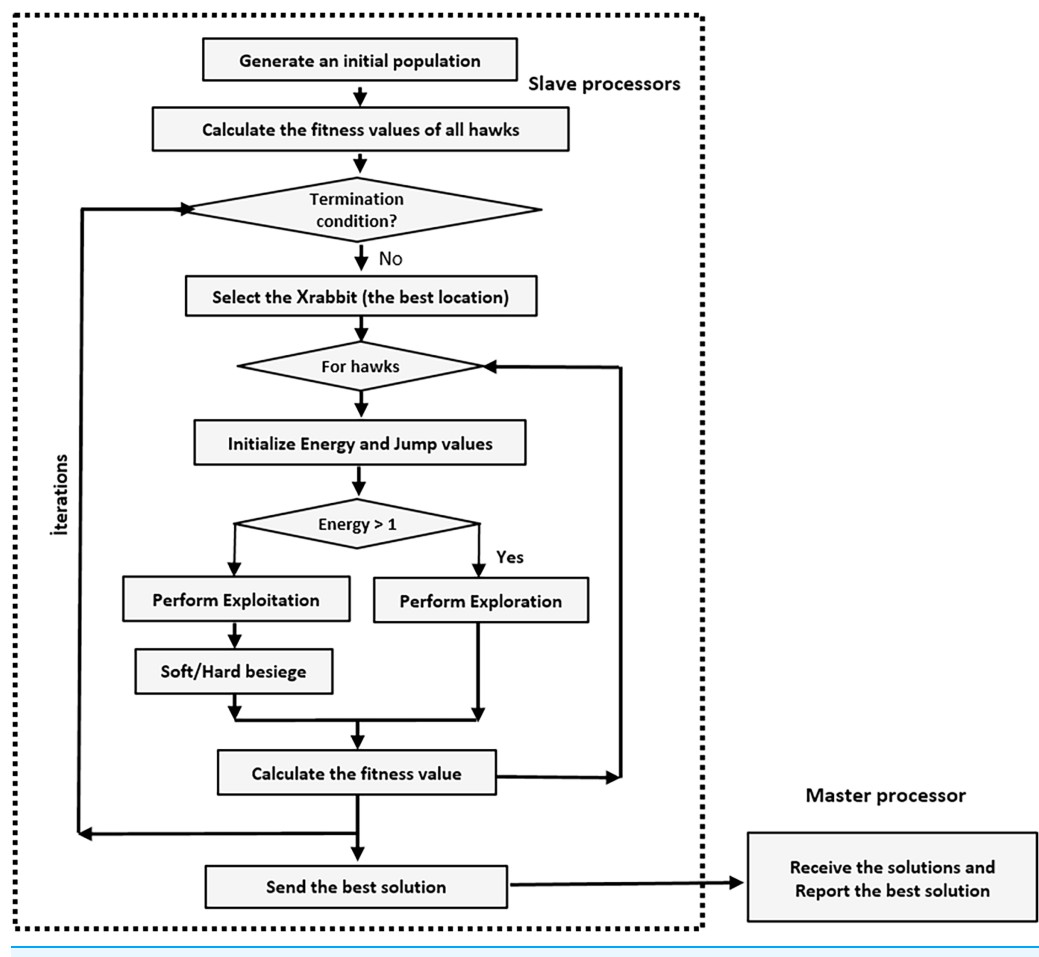

**Figure 4 The flowchart of the proposed PHHO-KNN algorithm**

*et al., 2003*), aiming to identify the K nearest data in the training set and find the label of the data. In KNN classification, the new point is assigned the most common label of its K neighbours. Selecting an appropriate value for K is crucial, as a small K value may lead to overfitting, while a large K value may lead to underfitting. KNN is an effective machine learning algorithm that performs well with small-sized datasets and has been applied successfully in various applications, such as image and speech recognition, text classification, and bioinformatics (*Batista & Monard, 2002*; *Larranaga et al., 2006*). However, the KNN algorithm can consume large amounts of time when dealing with large datasets. Figure 5 illustrates an example of the KNN classifier with K = 5, where point X belongs to the green class because it has more neighbours from this set.

We set the value of K to 5 in our proposed algorithm and performed experiments with K values ranging from 1 to 70 to investigate its effect on classification performance. The experiments were conducted on 34 features of the original COVID-19 dataset using 5-fold cross-validation, and each K value was tested 30 times. The results in Fig. 6 indicate that the best classification performance was achieved with K = 5. This experiment provided
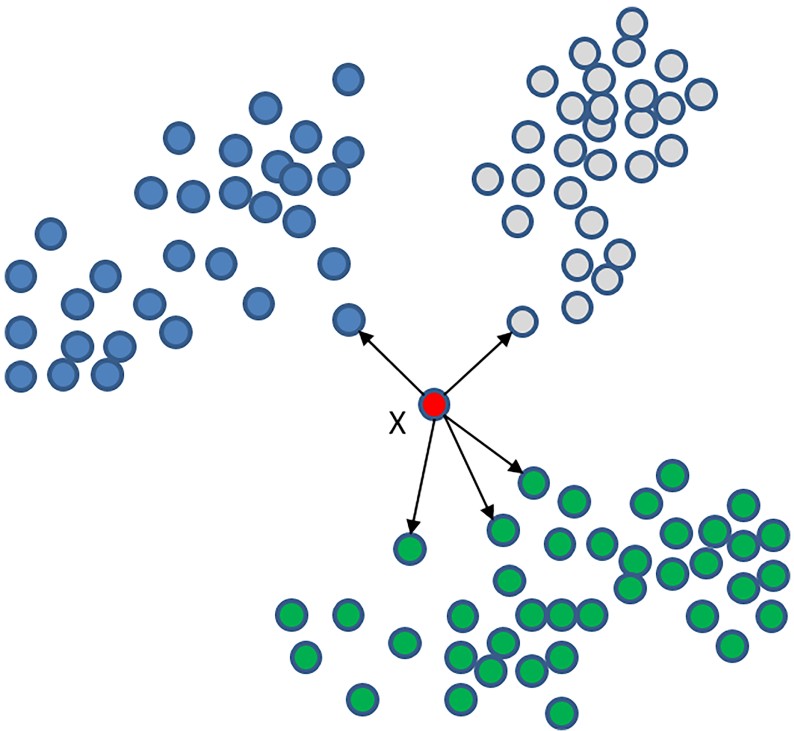

**Figure 5 KNN classifier with K = 5 for point X that is a member of the green class because it has more neighbours from this set.**

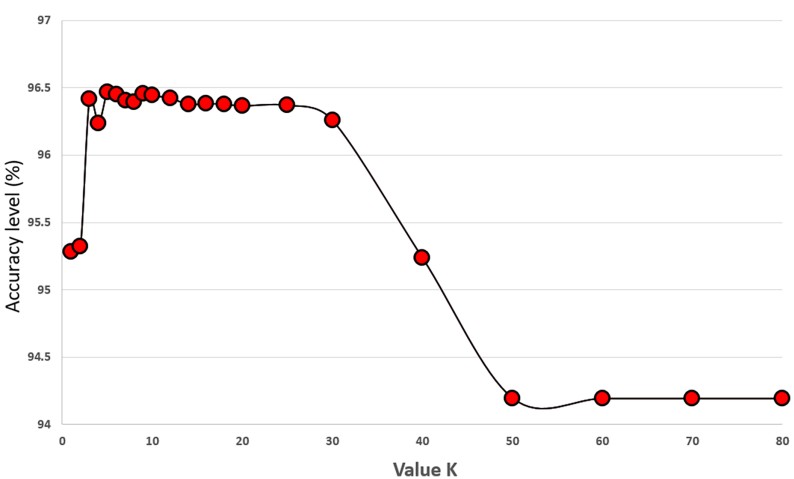

**Figure 6 Experiments showing the effect of K value (the number of neighbours) on the prediction accuracy performance of the KNN classifier.**

valuable insights into how the choice of K impacts classification performance, and setting it to five proved to be a critical step in improving the performance of our algorithm.

Manhattan distance is also known as L1 distance and is widely used in various machine learning algorithms. In addition to its effectiveness in feature selection studies, the Manhattan distance metric has advantages over other distance metrics in some applications. For instance, in recommendation systems, Manhattan distance can be used to

calculate the similarity between two users by measuring the distance between their preferences for different items (*Malkauthekar, 2013*). It is also used in computer vision for image processing, which calculates the distance between two images based on their pixel values. Another advantage of the Manhattan distance is that it is less sensitive to outliers than the Euclidean distance metric.

# EXPERIMENTAL SETUP AND EVALUATION OF THE RESULTS

This part of our study includes the experimental setup, COVID-19 Dataset details, proposed algorithm, and comparison with existing state-of-the-art algorithms. We used an AMD Opteron Processor 6,376 with a Multiprocessing Non-Uniform Memory Access architecture, consisting of four sockets with eight cores each, capable of running 64 threads simultaneously. Each node has eight processors and a Last Level Cache of 6 MB, with 64 GB of RAM distributed across eight nodes. The implementation was carried out using the C++ programming language and MPI libraries. To ensure accuracy, we employed five-fold cross-validation by randomly splitting the data into five partitions. The KNN was trained using four partitions and validated with the remaining partition.

## COVID-19 dataset

The original dataset is obtained from Kaggle (https://www.kaggle.com/datasets/sudalairajkumar/novel-corona-virus-2019-dataset), and it has missing and redundant values. Therefore, we preprocessed this version of the dataset (*Iwendi et al., 2020*; *Dokeroglu, Deniz & Kiziloz, 2022*).

We extracted all symptoms from the original dataset and categorized them into 24 features. Some of the symptoms included in our dataset are muscle pain, fatigue, sputum, headache, sore throat, thirst, runny nose, vomiting, pneumonia, cold, chills, cough, fever, sneezing, physical discomfort, nausea, reflux, abdominal pain, diarrhoea, loss of appetite, chest pain, difficulty breathing, flu, and joint pain. All of these symptoms were used as new features in our dataset. The original dataset combined these symptoms into columns using two or three values, with a value of one selected for each feature if the symptom was present. The number of days passed during the disease was also added as a new feature. The dataset consists of 1,085 rows (patients) and 34 features, with each symptom defined as a new feature. Integer encoding was employed for the text-based features.

We prepared an augmented dataset using the original COVID-19 dataset with 1,085 rows. The new augmented dataset has 1,192 rows, representing a 10% increase from the original dataset. The new rows were randomly selected, and no corrections were made to the patients' classification data. In this new augmented dataset, we reduced the age of the patients by 5% or made updates to the analysis data, which did not exceed 1%. We tested our proposed algorithm on both the original and augmented datasets and the following sections demonstrate how the augmented data increases the prediction accuracy of our proposed algorithm.

The most informative features (from the COVID-19 dataset) selected by our proposed algorithm are age, diff_sym_hos, from_wuhan, location, hosp_vis, pneumonia,

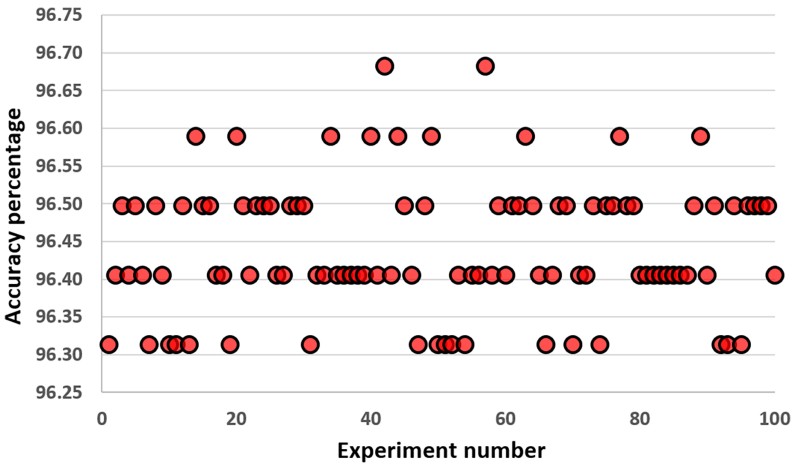

**Figure 7** **Results obtained with all features of the original COVID-19 datasets (100 samples are produced during the experiments).**

diff_sym_hos, fever, sym_on, vis_wuhan, diff_sym_hos, pneumonia, and difficulty in breathing.

## Setting the best number of neighbours (K value) for the KNN classifier

In this part, we performed experiments with different values of K. Figure 6 shows the results of our experiments. Increasing the number of K up to 30 performs well on the KNN classifier. Values larger than 50 harm the performance of the KNN classifier for the COVID-19 dataset. We select the value of K as five because it gives us the highest accuracy results during our experiments. These results are the average values of 30 runs from 1 to 70. Differences up to 2.23% are observed between the best and worst results. Figure 7 gives the results of all features of the original COVID-19 datasets (100 samples are produced during the experiments). The distribution of the results can be observed in this plot using the K value as five.

## Experiments with augmented COVID-19 dataset

In this section, we use an augmented COVID-19 dataset to enhance the quality of our results. Data augmentation techniques involve generating slightly modified copies of existing or synthetic data from the original dataset to expand the amount of data available (*Kaushik, Hovy & Lipton, 2019*). When training a machine learning model, data augmentation serves as a regularizer and helps in minimizing overfitting. It is closely related to data analysis oversampling. Our optimization uses a population size of 30 and runs for 500 generations, generating 15,000 new candidate solutions *via* operators and evaluating their fitness values during optimization. In a parallel computing environment with 64 processors, 64 distinct populations are distributed among these processors' memory, and 15,000 fitness evaluations are performed on each processor. This is a significant effort aimed at classifying the patients in our dataset. Our proposed algorithm performs a total of 960,000 fitness evaluations. We attempt to breed different individuals from the initial populations by seeding the randomized functions with the processors' ID
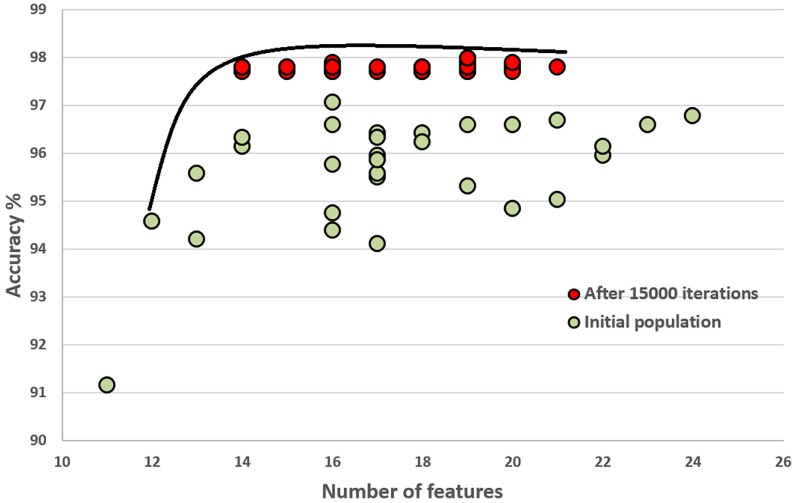

**Figure 8** Comparison of the initial population with the evolved population after 15,000 iterations on the original COVID-19 dataset.

numbers. Before parallelization, the execution time of the algorithm was 614.2 s. After implementing parallelization using 64 processors, the execution time increased slightly to 620.4 s. However, our goal was to improve the efficiency of the algorithm and process more fitness values. With a single processor, we were able to complete 15,000 fitness values, but with 64 processors, we were able to process 960,000 fitness values in nearly the same amount of time. This indicates an almost linear speedup in the optimization time of our algorithm.

## Initial population and how it improves through the generations

In this section, we compare the quality of our random initial population of solutions with the final population's quality. These experiments show that the initial population evolves after executing generations (iterations). The final population fits a better Pareto curve. Figures 8 and 9 compare the initial and final populations of the original COVID-19 and augmented COVID-19 datasets, respectively. The prediction accuracy results of the final populations are better in both cases for the original and augmented versions of the COVID-19 datasets. For the original and augmented COVID-19 datasets, the accuracy levels are improved by 2.14% and 2.95%, respectively. Two features for the augmented dataset reduce the number of features, while the average number of features is the same for the original COVID-19 dataset. These results are the average of 30 different executions of the experiments.

Figure 10 compares the quality of the final populations for the original and augmented COVID-19 datasets. Our proposed algorithm, PHHO-KNN, performs better with augmented data in terms of prediction accuracy (an improvement of 0.28% is observed). For our second objective, the average number of features used in the original and augmented datasets are 17.5 and 18.63, respectively. We have achieved an almost 45% reduction in the number of features used for classification.

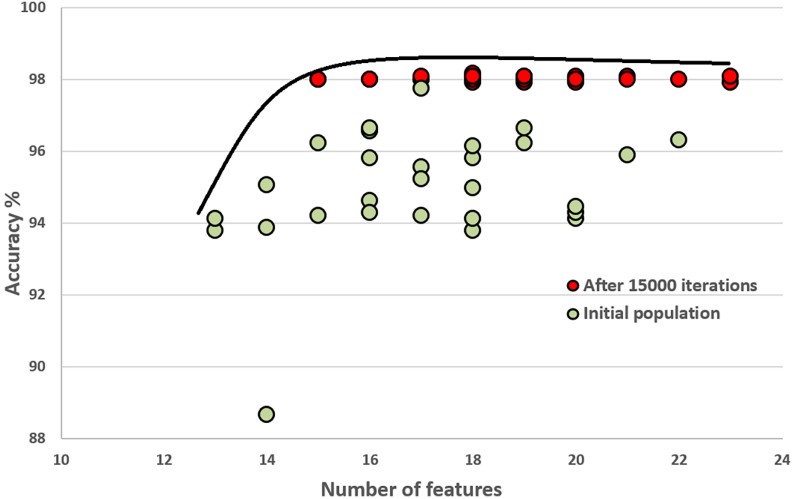

**Figure 9 Comparison of the initial population with the evolved population after 15,000 iterations on the original augmented COVID-19 dataset.**

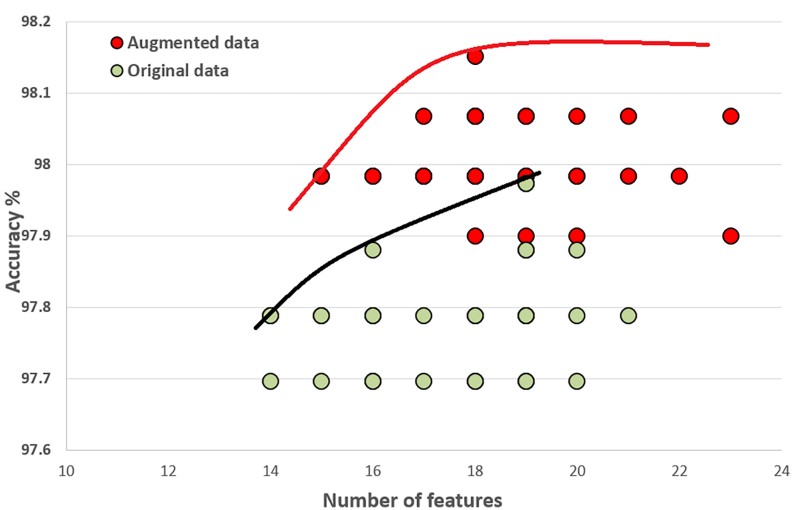

**Figure 10 Comparing original COVID-19 dataset results with augmented dataset of COVID-19.**

## Comparison with state-of-the-art algorithms

This section compares our solutions with state-of-the-art algorithms in the literature. Table 1 compares eight recent algorithms that have used the same COVID-19 dataset, all developed within the last 2 years. HLBDA (*Too & Mirjalili, 2021*), Boosted Random Forest (*Iwendi et al., 2020*), MHHO with ELM (*Dokeroglu, Deniz & Kiziloz, 2021*), LSRGNFM-LDC (*Mydukuri et al., 2022*), MG-ELM (*Dokeroglu, Deniz & Kiziloz, 2021*), MHHO with SVM, MHHO with Logistic Regression, and MHHO with Decision Trees are some of the best algorithms that have reported prediction accuracy values over 92.2%, although However, not all of them are multi-objective algorithms. We obtained the results from their related studies. Our proposed algorithm, PHHO-KNN, outperforms all the algorithms in the literature. The MG-ELM algorithm is also a parallel metaheuristic algorithm developed

**Table 1 Maximum accuracy values of the state-of-the-art algorithms on COVID-19 dataset.** Sorted by their accuracy values.

| Method | Accuracy (%) |
|---|---|
| HLBDA (*Too & Mirjalili, 2021*) | 92.21 |
| Boosted Random Forest (*Iwendi et al., 2020*) | 94.00 |
| MHHO with ELM (*Dokeroglu, Deniz & Kiziloz, 2021*) | 94.66 |
| LSRGNFM-LDC (*Mydukuri et al., 2022*) | 95.00 |
| MG-ELM (*Dokeroglu, Deniz & Kiziloz, 2021*) | 96.22 |
| MHHO with SVM (*Dokeroglu, Deniz & Kiziloz, 2021*) | 96.41 |
| MHHO with Logistic Regression (*Dokeroglu, Deniz & Kiziloz, 2021*) | 96.68 |
| MHHO with Decision Trees (*Dokeroglu, Deniz & Kiziloz, 2021*) | 97.61 |
| PHHO-KNN (our algorithm) | 97.88 |
| PHHO-KNN (with augmented data) | 98.15 |

**Table 2 Accuracy, Matthews correlation coefficient (MCC), precision, and recall values for the original and augmented COVID-19 datasets.**

| Value | Original dataset (%) | Augmented dataset |
|---|---|---|
| Accuracy | 97.88 | 98.15 |
| MCC | 0.85 | 0.90 |
| Precision | 73.25 | 81.96 |
| Recall | 1.0 | 1.0 |
| True positive (TP) | 63 | 100 |
| True negative (TN) | 999 | 1,070 |
| False positive (FP) | 23 | 22 |
| False positive (FN) | 0 | 0 |

using Java threads and was executed with eight threads. The MHHO algorithm provides the closest results to our solutions with the decision trees. Our results are 0.20% and 0.47% better than this algorithm's solutions for the original COVID-19 and its augmented version, respectively.

Table 2 presents the evaluation metrics for the performance of an algorithm on the original and augmented COVID-19 datasets. The accuracy values show that the algorithm has high accuracy on both datasets, with a slightly higher accuracy on the augmented dataset. This suggests that the algorithm is able to correctly classify a high percentage of the samples in both datasets. The Matthews correlation coefficient (MCC) (*Chicco & Jurman, 2020*) values are also high for both datasets, with a higher value on the augmented dataset. MCC is a measure of the quality of binary classifications, indicating a strong correlation between predicted and true labels. The precision values are substantially improved for the augmented dataset compared to the original dataset. This means that the algorithm has a higher proportion of true positive predictions than the augmented dataset's total positive predictions. The recall values are the same for both datasets, indicating that the algorithm

is able to identify all positive samples in both datasets. This suggests that the algorithm is able to avoid false negatives in both datasets. Finally, looking at the TP, TN, FP, and FN values, we can see that the number of TP is substantially higher in the augmented dataset compared to the original dataset, while the other values are similar between the two datasets. This indicates that the augmented dataset has more positive samples, which the algorithm is able to detect with high precision and recall values. Overall, the results show that the augmented dataset improved the algorithm's performance in terms of precision while maintaining high recall and accuracy values. This suggests that data augmentation techniques can be a useful tool in improving the performance of machine learning algorithms.

## CONCLUSIONS AND FUTURE WORK

This study presents a new robust parallel multi-objective HHO algorithm, PHHO-KNN, for predicting COVID-19 patient mortality. Parallel metaheuristics are powerful tools for effectively optimizing NP-hard problems with scalable abilities. Our algorithm uses the HHO metaheuristic for the feature selection layer and KNN as a classifier for the COVID-19 dataset. The PHHO-KNN algorithm can take the feature set of any classification problem dataset, including the output classes, and perform multi-objective optimization using feature selection. This provides a flexible framework for any classification problem, allowing for efficient and effective feature selection.

We observed that the number of fitness calculations increases with the number of processors in the computational environment, and the results can be improved by adding more processors. Our algorithm's results outperform state-of-the-art studies in the literature, achieving 98.15% accuracy with an augmented dataset. Additionally, on average, we reduce the number of features by 45%.

The primary limitation of our algorithm is the computation of fitness values for every new individual, which can take a significant amount of time when the number of instances and features (data size) is large. Generating distinct individuals may not always be feasible for all slave nodes. While certain mechanisms can control this, they may take time and may not be scalable as we increase the number of processors in our environment.

Selecting the appropriate number of processors, and to optimizing the number of fitness calculations are interesting parameters to be set for the parallel metaheuristic algorithms. Since the possibility of obtaining better results is increased using as many processors as possible we used the maximum number of processors during our experiments.

In the future, we intend to incorporate deep learning classifiers with our parallel multi-objective HHO algorithm to improve its performance further. Hyper-heuristics is an attractive field of research, and applying state-of-the-art hyper-heuristic frameworks to this domain can make an important contribution. Furthermore, we plan to examine newly published COVID datasets to expand the scope of our research. This field has great potential to classify patients with various diseases such as Parkinson's. We consider directing our research area to classify other hard-to-identify kinds of diseases. According to the no-free lunch theorem, new opportunities for better solutions with new metaheuristics will always exist. In particular, quantum-based metaheuristic models have

been popular recently, and they report significant improvements in this field. These topics can be considered open research questions of this domain.

### Funding
The author received no funding for this work.

### Competing Interests
The author declares that they have no competing interests.

### Author Contributions
- Tansel Dokeroglu conceived and designed the experiments, performed the experiments, analyzed the data, performed the computation work, prepared figures and/or tables, authored or reviewed drafts of the article, and approved the final draft.

### Data Availability
The raw data and code are available in the Supplemental Files.

### Supplemental Information
Supplemental information for this article can be found online at http://dx.doi.org/10.7717/peerj-cs.1430#supplemental-information.

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
