# Peer review of "A new parallel multi-objective Harris hawk algorithm for predicting the mortality of COVID-19 patients"

_PeerJ Computer Science, doi:10.7717/peerj-cs.1430_

## Round 0.1 · original submission · Major Revisions

The reviewers assessed the article and noticed some important aspects to improve. I have the sense there could be some relevant result related to COVID19 mortality, but there are actually too many issues in the current article version. The authors are invited to address the points of the reviewers and to prepare a new article version.

Reviewer 1 ·

Basic reporting

The authors propose a multi-objective HHO algorithm for predicting mortality in COVID-19 patients based on their symptoms. However, the whole paper seems rather tedious. On the one hand, the paper is not innovative enough. The authors simply extend the HHO into a multi-objective version, which is not novel enough, and this has already been published in the literature [1]. On the other hand, the layout, description, and experiments of the article need to be rationalized.

1. Jangir, P., A.A. Heidari, and H. Chen, Elitist non-dominated sorting Harris hawks optimization: Framework and developments for multi-objective problems. Expert Systems with Applications, 2021. 186: p. 115747.

The specific comments are as follows:

1. Please describe in sufficient detail the specific design specifics of the parallel multi-objective HHO algorithm, which is of interest to most readers, rather than a single description of the HHO.

Experimental design

1. The design of the objective function often affects the performance of multi-objective feature selection algorithms when dealing with multi-objective feature selection problems. Therefore, it is recommended that the author describes in detail the objective function used in this study.

2. The author uses an iterative approach to compare with the state-of-the-art multi-objective algorithms, which is unfair. Therefore, it is recommended that the author uses an evaluation approach for comparison, which often yields interesting results.

Validity of the findings

1. Between the concluding chapters, it would be useful for the author to have a separate discussion section to provide an in-depth analysis of the experiments carried out in this paper, where are the strengths and weaknesses compared to existing methods? What specific areas of deficiency still exist? This would facilitate a better tour for the reader.

2. I noted the brief paragraph addressing future work in the Conclusion Section. However, this failed to consider open research questions (ORQ) identified in the study along with related directions for future research. This is important and must be provided.

3. In all such research, an important consideration lies in the practical managerial significance and practical application of the proposed method. A suitable discussion on this topic is required to show the potential applications for the model and the ability to generalize to multiple domains.

4. The limitations of this work should be added to the conclusions section.

5. References should include DOI numbers, please reorganize and adjust them by the author.

Reviewer 2 ·

Basic reporting

The authors of this article have proposed a new parallel multi-objective HHO algorithm for predicting the mortality of COVID-19 patients according to their symptoms. An augmented data version of the COVID-19 dataset is also generated and analyzed to improve the quality of the solutions. However, I would like to suggest some points mentioned below which are very much necessary to address.
1. The motivation behind the consideration of HHO algorithm for feature selection should be mentioned clearly.
2. The author’s contribution should be listed in the introduction section.
3. The majority of works explained in literature review section are related to COVID-19 prediction. As you are proposing an improved multi-objective FS technique you should mention some existing works related to FS using multi-objective evolutionary approaches.
4. In proposed work there is a line “There are two layr in our proposed algorithm”. Correct the spelling of layer.
5. Write full form of ‘pf’ in its first appearance.
6. Why parallelization is required for multi-objective HHO in FS, should be explicitly mentioned before the proposed algorithm with proofs from existing works.

Experimental design

1. A block diagram of the proposed architecture is highly appreciated.
2. Table 1 lists only the accuracy value. However, you are proposing a multi-objective algorithm means you should mention the accuracy value with respect to the number of features for better comparison.
3. Pareto front is the best method for visualizing the output of any multi-objective algorithm. But I didn’t find it anywhere in the paper.
4. The author should compare the performance of the proposed algorithm with the existing works in terms of some multi-objective performance indicators (GD, IGD, HV etc…).

Validity of the findings

1. The performance of the proposed method should be verified statistically.
2. As the author is proposing parallel multi-objective HHO, they should mention the execution time required.
3. Complexity analysis not done.
4. The strengths and weaknesses of the proposed work should be mentioned.
5. Future work should be listed as per the limitations of the proposed work.

Additional comments

Improve the quality of writings and cite some recent papers in multi-objective HHO for Feature selection.

·

Basic reporting

Clear and unambiguous, professional English used throughout.

Experimental design

There is a need to elaborate more on the methodology and data.

Validity of the findings

The results need to be further explained

Additional comments

-

---

## Round 0.2 · Minor Revisions

The reviewers appreciated the article but raised a number of minor points that need to be addressed. I therefore invite the authors to prepare a new version of the manuscript taking into account the comments made by the reviewers and resubmit the paper.

Reviewer 1 ·

Basic reporting

A new parallel multi-objective HHO algorithm is proposed to predict the mortality of patients with COVID-19. The author used the real world COVID-19 dataset from Kaggle to conduct experiments, and compared it with the most advanced meta heuristic packaging algorithm. The conclusion mentions that the algorithm has strong adaptability and can be easily applied to the classification of other datasets, and the results can be improved by increasing the number of processors. However, I think some of the descriptions in this article are still a bit vague. Next, let me enumerate these problems:

1. The article can provide a clearer explanation of why HHO was chosen as the solution to the problem, and how HHO can help solve multi-objective optimization problems.

2. It can provide more practical information about parallel meta heuristics, such as how to select the appropriate number of processors, and how to optimize the number of fitness calculations.

3. The author stated that the PHHO-KNN algorithm has strong adaptability, but did not provide a specific explanation.

4. I suggest the authors make a comprehensive investigation of the optimization methods in the literature in the introduction part and give the analysis to the existing works such as (https://doi.org/10.1016/j.compbiomed.2021.104968,
https://doi.org/10.1016/j.compbiomed.2021.104984,
https://doi.org/10.1016/j.compbiomed.2021.104712,
https://doi.org/10.1007/s42235-022-00297-8, https://doi.org/10.1016/j.compbiomed.2021.104558) to make the whole work more in-depth.

5. The author should describe more relevant research achievements in recent years in the introduction. I suggest the authors to introduce some recently proposed meta-heuristics such as the slime mould algorithm (SMA), Runge Kutta method (RUN), colony predation algorithm (CPA), weighted mean of vectors (INFO) and rime optimization algorithm(RIME)to make the paper get more readers.

Experimental design

no comment

Validity of the findings

no comment

Additional comments

no comment

Reviewer 2 ·

Basic reporting

No comments

Experimental design

No comments

Validity of the findings

Cross check the complexity analysis.
Author should mention the execution time before and after parallelization.
It will be better if the performance of the proposed method will be verified statistically.

Additional comments

Once again check the grammar, spellings, and clarity of the figures.

·

Basic reporting

Clear and unambiguous, professional English used throughout.
Literature references, sufficient field background/context provided

Experimental design

Methods described with sufficient detail & information to replicate

Validity of the findings

Conclusions are well stated, linked to original research question & limited to supporting results.

Additional comments

-

---

## Round 0.3 · Minor Revisions

The authors clearly addressed the points raised by the reviewers, improving the article profoundly.
However, I reviewed the article and I noticed that the results are based on accuracy, which is a misleading measure ( https://doi.org/10.1186/s12864-019-6413-7 ). I therefore invite the authors to repeat the tests and measure their results through Matthews correlation coefficient (MCC), precision, negative predictive value, and comment on these results in the article.

Reviewer 1 ·

Basic reporting

From the response letter, the paper has been well revised, and the current version of the manuscript is acceptable for publication.

Experimental design

no comment

Validity of the findings

no comment

Additional comments

no comment

---

## Round 0.4 · accepted · Accept

The author address all the comments raised by me and by the reviewers and therefore I recommend this article for publication.